# Therapeutic Potential of a Combination of Magnesium Hydroxide Nanoparticles and Sericin for Epithelial Corneal Wound Healing

**DOI:** 10.3390/nano9050768

**Published:** 2019-05-19

**Authors:** Noriaki Nagai, Yoshie Iwai, Saori Deguchi, Hiroko Otake, Kazutaka Kanai, Norio Okamoto, Yoshikazu Shimomura

**Affiliations:** 1Faculty of Pharmacy, Kindai University, 3-4-1 Kowakae, Higashi-Osaka, Osaka 577-8502, Japan; 1311610058c@kindai.ac.jp (Y.I.); 1111610121m@kindai.ac.jp (S.D.); hotake@phar.kindai.ac.jp (H.O.); 2Department of Small Animal Internal Medicine, School of Veterinary Medicine, University of Kitasato, Towada, Aomori 034-8628, Japan; kanai@vmas.kitasato-u.ac.jp; 3Okamoto Eye Clinic, 5-11-12-312 Izumicho, Suita, Osaka 564-0041, Japan; eyedoctor9@msn.com; 4Department of Ophthalmology, Fuchu Hospital, 1-10-17 Hikocho, Izumi, Osaka 594-0076, Japan; y_shimomura@seichokai.or.jp

**Keywords:** sericin, nanoparticle, corneal wound, magnesium hydroxide, ophthalmic formulation

## Abstract

We previously found the instillation of sericin to be useful as therapy for keratopathy with or without diabetes mellitus. In this study, we investigated whether a combination of solid magnesium hydroxide nanoparticles (MHN) enhances epithelial corneal wound healing by sericin using rabbits, normal rats and type 2 diabetes mellitus rats with debrided corneal epithelium (ex vivo and in vivo studies). Ophthalmic formulations containing sericin and MHN (N-Ser) were prepared using a bead mill method. The mean particle size of the N-Ser was 110.3 nm at the time of preparation, and 148.1 nm one month later. The instillation of N-Ser had no effect on the amount of lacrimal fluid in normal rabbits (in vivo), but the MHN in N-Ser was found to expand the intercellular space in ex vivo rat corneas. In addition, the instillation of N-Ser increased the phosphorylation of Extracellular Signal-regulated Kinase (ERK)1/2, a factor involved in cell adhesion and cell proliferation in the corneal epithelium, in comparison with the instillation of sericin alone. The combination with MHN enhanced epithelial corneal wound healing by sericin in rat debrided corneal epithelium (in vivo). This study provides significant information to prepare potent drugs to cure severe keratopathy, such as diabetic keratopathy.

## 1. Introduction

The epithelium forms the outermost layer of the cornea, and plays an essential role in preserving corneal clarity and protecting the eye against pathogen invasion. Thoft and Friend [1] proposed the XYZ theory, according to which cells move initially through the deeper layers of the epithelium (X), becoming more differentiated as they approach the central region of the cornea, and elevate to a superficial stage (Y) until they complete their life cycle between 5 and 7 days later, and then detach (Z). Due to the high resistance of the junctions between epithelial cells, the cornea acts as a physical barrier that protects the eye [2,3]. Following injury to the cornea, a highly coordinated repair process involving cellular proliferation, migration, differentiation and death occurs in the corneal epithelium. The process of epithelial corneal wound healing involves the shrinkage of the corneal epithelial cells, with the corneal wound surface covered by differentiating cells (Y) after the loss of the epithelial cells on the corneal surface (Z). Furthermore, cell proliferation provides for the remodeling and rebuilding of the tissue (X) [1,4,5,6]. Small injuries to the corneal epithelium are thus restored spontaneously. However, for the treatment of major injuries, a way to increase corneal re-epithelialization is needed that prevents corneal neovascularization and trauma [7]. Clinically, there has been great progress in the treatment of corneal wounds, with supportive measures in the forms of antibiotics, lubricants, tarsorrhaphy, and bandage contact lenses being used for the conventional treatment of corneal wounds. However, wound healing in the case of corneal disease or severe damage remains challenging [8].

Sericin is a natural glue-like protein isolated from Bombyx mori silkworm cocoons. Sericin is reported to be a mitogenic factor that promotes better cell attachment [9,10] and cell proliferation [11,12,13], as well as better developmental competence of mammalian zygotes [14]. Experimentally, it is used in the culture of hybridomas, T-lymphocytes, endothelial cells, human marrow stromal cells [15,16], human skin fibroblast cells [17], and Sf9 insect cells [18]. Previously, we reported that sericin solutions activate the Mitogen-Activated Protein Kinase/Extracellular Signal-regulated Kinase (MAPK/ERK) pathway, and enhance cell proliferation resulting in an increase in epithelial corneal wound healing [19]. These results show that sericin has the potential to act as an epithelial corneal wound-healing drug. However, delivery of the drug is challenging as the components in eye drops are rapidly cleared following instillation, and excreted into the mouth via the nasolacrimal duct [20]. Therefore, the design of a system to maintain high sericin levels on the cornea after instillation is important.

Recently, we reported that the co-instillation of magnesium hydroxide (MH) nanoparticles (MHN) increases the ratio of the intercellular space in corneal epithelium (the intercellular area/ the intercellular plus epithelium area), resulting in an increase in intracellular uptake and corneal penetration of dissolved drugs such as timolol maleate and carteolol [21,22]. Thus, the co-instillation of MHN may be useful for enhancing the therapeutic effects of ophthalmic formulations by prolonging the time drugs remain on the cornea. In this study, we demonstrate that the co-instillation of MHN with sericin increases the rate of epithelial corneal wound healing of debrided corneal epithelium in rats.

## 2. Materials and Methods

### 2.1. Animals

Japanese albino rabbits (male, 2.5–3.0 kg, *n* = 8), 7 week-old Wistar rats (220–250 g, *n* = 58) and 40 week-old Otsuka Long-Evans Tokushima Fatty (OLETF) rats (body weight, 590–660 g; plasma glucose levels, 201.3–244.7 mg/dL, *n* = 31), a model of type 2 diabetes mellitus, were used in this study. The rabbits, Wistar rats, and OLETF rats were obtained from Shimizu Laboratory Supplies Co., Ltd. (Kyoto, Japan), Kiwa Laboratory Animals Co., Ltd. (Wakayama, Japan), and Japan SLC Inc. (Shizuoka, Japan), respectively. All experiments were performed in accordance with the guidelines for the Association for Research in Vision and Ophthalmology (ARVO). In addition, the experiments using rats were approved on 1 April 2013 (project identification code, KAPS-25-003) and 2015 (project identification code, KAPS-27-017) by the Pharmacy Committee Guidelines for the Care and Use of Laboratory Animals in Kindai University.

### 2.2. Chemicals

All other chemicals used were of the highest purity commercially available. Pure Sericin™ (30 kDa), isoflurane, MH microparticles (MHP), mannitol and the Magnesium B test kit were purchased from Wako Pure Chemical Industries, Ltd. (Osaka, Japan). The BD Micro-Sharp™ (blade 3.5 mm, 30°) and disposable dermatological skin punch (BIOPSY PUNCH) were provided by Becton Dickinson (Fukushima, Japan) and Kai Industries Co., Ltd. (Gifu, Japan), respectively. Methylcellulose of type SM-4 (MC) was supplied by Shin-Etsu Chemical Co., Ltd. (Tokyo, Japan). Sterilized Tear Production Measuring Strips were purchased from AYUMI Pharmaceutical Corporation (Tokyo, Japan). Benoxil (0.4%) and benzalkonium chloride (BAC) were obtained from Santen Pharmaceutical Co., Ltd. (Osaka, Japan) and Kanto Chemical Co., Inc. (Tokyo, Japan), respectively. Fluorescein was provided by Alcon Japan (Tokyo, Japan). Dulbecco’s modified Eagle’s medium/Ham’s F12 (DMEM/F12), heat-inactivated fetal bovine serum, penicillin, and streptomycin were purchased from GIBCO (Tokyo, Japan), and Cell Count Reagent SF was obtained from Nacalai Tesque (Kyoto, Japan). Sterilized Tear Production Measuring Strips were provided by SHOWA YAKUHIN KAKO Co., Ltd. (Tokyo, Japan). Phospho-p44/42MAPK (Erk1/2)(Thr202/Tyr204)(D13.14.4E) XP^®^Rabbit mAb and P44/42MAPK(Erk1/2) antibody were purchased from Cell Signaling Technology, Inc. (Danvers, MA, USA). Affinity purified secondary antibodies against rabbit IgG (H and L) adsorbed against rat, horse, bovine, and human, and conjugated to Horseradish Peroxidase (secondary rabbit IgG) were obtained from Promega (Madison, WI, USA).

### 2.3. Preparation of N-Ser

The 1% sericin solution and nanoparticles (MHN) were prepared using a bead mill method reported by previous study [21,22,23,24]. The sericin ophthalmic formulations (Sericin/MH fixed combination) containing MHP (P-Ser) or MHN (N-Ser) were prepared by mixing the sericin solution and MH formulations (MHP and MHN). Figure 1 shows the scheme in the preparation process of N-SER. The final concentrations in the P-Ser and N-Ser were as follows: sericin 1%, MH 0.01%, MC 0.5%, BAC 0.005%, mannitol 0.5%, pH8.5, isotonization, sterilized). The size of the MH particles N-Ser was determined by both a laser diffraction particle size analyzer SALD-7100 (Shimadzu Corp., Kyoto, Japan) and dynamic light scattering NANOSIGHT LM10 (Quantum Design Japan, Tokyo, Japan). The Atomic Force Microscope (AFM) images of MHNand N-Ser were provided by an SPM-9700 (Shimadzu Corp., Kyoto, Japan) [21,22].

### 2.4. Stability and Solubility of MH Particles N-Ser

The P-Ser and N-Ser formulations were preserved in the dark for 1 month at 20 °C, and the stability were evaluated [21,22]. The MH solubility in saline and lacrimal fluid of rabbits were determined using a Magnesium B test kit according to the manufacturer’s instructions. Briefly, the samples were mixed with solution containing glucokinase, ATP, glucose and nicotinamide adenine dinucleotide phosphate, and incubated for 3 min, and measured the Abs (340 nm) by UV-2200 (Shimadzu Corp., Kyoto, Japan).

### 2.5. Measurement of Lacrimal Fluid after Instillation of N-Ser

The amount of lacrimal fluid was determined using the Sterilized Tear Production Measuring Strips (Schirmer’s test). In this study, we used normal rabbits in study for long-term instillation. The rabbits were applied the general 30 µL instillation (twice a day, 9:00 a.m. and 7:00 p.m.) for 1 month (total 60 times, 5 rabbits/per condition).

### 2.6. Observation of the Rat Corneal Surface

The rats were euthanized by the excessive pentobarbital, and 5 samples/group were used for ex vivo testing of intercellular space in the corneal epithelium. Corneas were removed from 7 week-old rats, and treated with saline, vehicle, P-Ser, or N-Ser (15 min, 20 °C). The corneal epithelium were stained by the fluorescein to discern the area in intercellular and epithelium, and observed by a phase contrast microscope, and the ratio of the corneal intercellular space (the area in intercellular/area in intercellular and epithelium, %) was analyzed using Image J software in the obtained picture (1 × 10^4^ µm^2^).

### 2.7. Measurement of Epithelial Corneal Wound Healing in Rats

Rats with corneal epithelium-debridement wounding were prepared according to our previous study [25,26]. TRC-50X (Topcon, Tokyo, Japan) were used to monitor the changes in the corneal wound area, and analyzed by Image J [25,26]. The rat with corneal abrasion is acute model. Therefore, 30 µL of saline or an ophthalmic formulation containing 1% sericin were multi instilled into the rat eyes five times a day (9:00 a.m., 0:00 p.m., 3:00 p.m., 6:00 p.m. and 9:00 p.m.) after corneal abrasion. Epithelial corneal wound healing (%) and the epithelial corneal wound healing rate constant (*k*_H_, h^−1^) were analyzed by the following Equations (1) and (2):Epithelial corneal wound healing (%) = (wound area 0 h − wound area 12–72 h)/wound area 0 h × 100(1)
(2)$$H_{t}=H_{\infty }\left(1-e^{-k_{H}t}\right)$$
where *t* is time after corneal abrasion, and *H*_∞_ and *H*_t_ are the percentages of epithelial corneal wound healing (%) at time *∞* and *t*, respectively.

### 2.8. Western Blot Analysis

The corneas were removed from rats euthanized by the excessive pentobarbital, and the proteins in the cornea were extracted according to our previous study [19]. Total protein (10 µg) was separated in a 10% polyacrylamide sodium dodecyl sulfate (SDS) gel, and transferred to polyvinylidene difluoride membranes. The phosphorylation of ERK1/2 (pERK) was probed with phospho-p44/42MAPK (Erk1/2)(Thr202/Tyr204)(D13.14.4E) XP^®^ rabbit mAb (1:1000 dilution, 10 h, 4 °C). After washing with Tris-buffer containing TEIRON X-100, the membranes were incubated with secondary anti-rabbit IgG (1:4000 dilution, 1 h, room temperature), treated with 1 ml Super Signal^®^ and the bands were detected using IQuant 400 for pERK. After that, the membranes were washed with stripping buffer (1 h), and the sites on the membranes were blocked with 5% non-fat dry milk in Tris-buffer (20 mM Tris-HCl, and 500 mM NaCl, pH 7.5), and used to measure the expression of ERK proteins. ERK was probed with P44/42MAPK(Erk1/2) antibody (1:1000 dilution, (10 h, 4 °C). Then, secondary anti-rabbit IgG was added, and incubated (1:4000 dilution, 1 h, room temperature). The band intensity of pERK1/2 phosphorylation was quantified by the Image J (NIH, USA), and represented as the ratio of control (non-treated groups).

### 2.9. Statistical Analysis

The sample numbers (*n*) are shown in the table and figure legends. Student’s *t*-test and ANOVA followed by Dunnett’s multiple comparison were used for statistical analysis (*P* < 0.05). The data from the SALD-7100 are expressed as the mean ± standard deviation (S.D.); other data are expressed as the mean ± standard error (S.E.) of the mean.

## 3. Results

### 3.1. Design of Ophthalmic Formulations Containing Sericin and MHN

We previously reported two findings: one is that the MHN expands the intercellular space in the cornea [21,22], and other is that sericin enhances the epithelial corneal wound healing [25,26]. In this study, we demonstrated the effect of MHN in combination with sericin on the therapeutic effect of sericin in corneal wounding. Figure 2 shows the particle size of MH in N-Ser. The size of the MH in P-Ser was 4.83 ± 0.31 µm (Figure 2A), and the particles were milled using the bead mill treatment. The size of the MH particle in N-Ser by NANOSIGHT LM10 (Figure 2C) was 110.3 ± 4.9 nm, and the size was similar to MHN (Figure 2B). In addition, we investigated the shape of the MH particles in N-Ser by AFM imaging, which showed the MH particles after bead mill treatment to be in the nano-size, and the shape was like a spherical shape (Figure 2F). Next, we measured the particle size frequency and shape of the MHN in N-Ser 1 month after preparation, since aggregated MH particles lose the characteristics of nanomaterials, such as the increase in ratio of intercellular space in the cornea. We measured the size and shape of the MH particles in the N-Ser using the NANOSIGHT LM10 (Figure 2C) and AFM imaging (Figure 2F,G), respectively. The mean particles size shifted to 148.1 ± 10.7 nm from 110.3 ± 4.9 nm, and the presence of three populations of beads (around 80, 160, and 240 nm) were observed, although the MH particles in N-Ser were still in the nano-order size range, and the shape of the MH particles 1 month after preparation was also similar to that immediately after preparation.

### 3.2. Therapeutic Effect of N-Ser in the Treatment of Epithelial Corneal Wound Healing

When investigating thetherapeutic potential of N-Ser for the treatment of epithelial corneal wound healing, it is also important to elucidate the state of the MH particles in N-Ser following instillation. Therefore, we measured the solubility of MH in lacrimal fluid (Figure 3A), and found it to be 3.71 ± 0.76 µg/mL, with the ratio of solid to dissolved MH of 96.29:3.71. This shows that the MH particles in N-Ser remain in the solid state on the corneal surface after instillation. In addition, we measured the MH particle size in lacrimal fluid using the NANOSIGHT LM10 (measurement conditions: lacrimal fluid: N-Ser = 1:1), and found it to be 146.2 ± 8.7 nm. Furthermore, we looked at whether repetitive instillation of N-Ser affects the corneal condition in rabbits, and found no effect on the amount of lacrimal fluid (Figure 3B), the ratio of the intercellular space in corneas instilled with N-Ser was higher than in corneas instilled with vehicle or P-Ser (Figure 3C). Next, the therapeutic effect of N-Ser in the treatment of epithelial corneal wound healing was evaluated in rats without (Wistar rats, Figure 4) or with (OLETF rats, Figure 5) diabetes mellitus. Wound areas were as follows: Wistar rat (saline, 10.2 ± 0.56 mm^2^; sericin, 9.9 ± 0.42 mm^2^; P-Ser, 10.6 ± 0.47 mm^2^; N-Ser, 10.5 ± 0.49 mm^2^) and OLETF rat (saline, 10.8 ± 0.51 mm^2^; sericin, 10.9 ± 0.53 mm^2^; P-Ser, 10.5 ± 0.46 mm^2^; N-Ser, 11.0 ± 0.49 mm^2^). In the Wistar rats instilled with saline, the corneal wounds were cured by approximately 15% and 65% healing at 12 and 24 h after abrasion, respectively. The corneal wounds were almost entirely healed after 36 h (Figure 4). On the other hand, the levels of epithelial corneal wound healing in OLETF rats was lower than in Wistar rats, with complete healing only after 72 h (Figure 5). Table 1 shows the effects of the ophthalmic formulations containing sericin on the epithelial corneal wound healing constant (*k*_H_) in Wistar and OLETF rats. The *k*_H_ in both Wistar and OLETF rats were enhanced by the instillation of sericin by 1.6- and 1.5-fold, respectively, in comparison with saline. Although the *k*_H_ values in rats instilled with P-Ser were similar to those instilled with sericin, the *k*_H_ values in rats instilled with N-Ser were significantly higher than in rats instilled with sericin. The *k*_H_ values in Wistar (1.3-fold) and OLETF rats (1.9-fold) instilled with N-Ser were higher than in rats instilled with sericin. 

### 3.3. Effect of the Combination of Sericin and MHN on ERK Signaling

It has been reported that sericin increases cell adhesion and cell proliferation via ERK1/2, resulting in an increased rate of epithelial corneal wound healing [19]. Therefore, we compared changes in ERK1/2 activation between rat instilled with sericin and N-Ser (Figure 6). The ERK1/2 phosphorylation in the corneas of rats instilled with N-Ser was higher than in those instilled with sericin, and the band intensity of pERK1/2 phosphorylation in the sericin and N-Ser were 1.8 ± 0.3 (*n* = 4) and 2.7 ± 0.4 (*n* = 5), respectively (control, 1.0 ± 0.3, *n* = 4).

## 4. Discussion

It is important to design treatments for epithelial corneal wound healing, since non-healing corneal wounds cause visual impairment in the setting of corneal diseases [27]. In this study, we designed ophthalmic formulations containing sericin and MHN (N-Ser), and found that the instillation of N-Ser provides useful therapy for keratopathy with or without diabetes mellitus.

Our previous study showed that almost all of the MHN (approximately 100 nm) were delivered to the stomach via the nasolacrimal duct after instillation, since no change in the Mg^2+^ levels in the aqueous humor of rabbits treated with MHN dispersions were observed [21]. Furthermore, we also reported that MHN in the range of 30–300 nm enhance the ratio of the corneal intercellular space, and increase the intracellular uptake and corneal penetration of dissolved drugs by co-instillation [21,22]. Therefore, we prepared ophthalmic formulations containing sericin and MHN by the bead mill method, and determined by the dynamic light scattering method, and AFM imaging (Figure 2). The particle size of the MH to be approximately 30–300 nm (Figure 2), and. the particle size frequency is also approximately 30–300 nm 1 month after preparation. Moreover, the shape of the milled-MH in the presence of sericin were not changed 1 month after preparation (Figure 2), and 96.29% of the MH remained as solid particles when instilled in lacrimal fluid (Figure 3A). In addition, the instillation of N-Ser into rabbit eyes had no effect on the amount of lacrimal fluid (Figure 3B), and expanded the intercellular space in the cornea (Figure 3C). Next, we demonstrated the therapeutic effects of N-Ser instillation. Corneal epithelium debridement is an ideal model in which to study reepithelialization and enhanced wound healing [28]. Using this model, we evaluated the rate of epithelial corneal wound healing following the instillation of N-Ser. The epithelial corneal wound healing rate in Wistar rats instilled with N-Ser was higher than following the instillation of sericin (Figure 4). These results showed combining MHN with sericin enhanced the therapeutic effect of sericin.

We also investigated the potential of N-Ser instillation for the therapy of diabetic keratopathy. The delayed epithelial wound healing is often observed in the patients with diabetic keratopathy [29], and delayed epithelial wound closure may be related with sight-threatening complications, including microbial keratitis stromal opacification, and surface irregularities [30]. It was known that 47–64% patients with diabetes mellitus suffer from keratopathies [31], and, furthermore, Gao et al. [32] reported that as many as 73.6% patients of diabetes mellitus endure the corneal complications, such as endothelial dystrophy, pannus, punctate keratopathy and corneal ulcers. The conventional treatment regimens do not show enough action against diabetic corneal complications, termed diabetic keratopathy [31,32], and are currently treated with bandage contact lens and antibiotics, lubricants, and tarsorrhaphy, all of which try to make a more favorable environment for epithelial corneal wound healing [33]. Nevertheless, these measures may be inadequate and ineffectiveat enhancing re-epithelialization in diabetes mellitus [33]. In our study, to demonstrate the potential of N-Ser instillation for the therapy of diabetic keratopathy, we chose type 2 diabetes mellitus model OLETF rats. These rats exhibit hyperglycemia and hyperinsulinemia at 40 weeks of age as a result of peripheral insulin resistance and islet cell hyperplasia [34,35,36,37]. The epithelial corneal wound healing rate in 40 weeks-old OLETF rats is delayed in comparison with control (normal) rats [25,38], and the changes in the biological characteristics of OLETF rats were similar to those in human patient with type 2 diabetes mellitus. Therefore, we measured the epithelial corneal wound healing rate by sericin with or without MHN in this study. Although the epithelial corneal wound healing rate in P-Ser-instilled rats were not different than that in sericin-instilled rats, the epithelial corneal wound healing in the N-Ser-instilled rats were significantly higher 12–60 h after corneal abrasion (Figure 5). These results suggest that N-Ser is useful for the therapy of epithelial corneal wound. In the evaluation of usefulness of N-Ser, it is important to compare the epithelial corneal wound healing between this data and previous result using sericin [25,26]. However, the corneal wound healing rate in N-Ser (containing 1% sericin) was lower than that in 10% sericin alone 12 h after corneal epithelial abrasion [25]. It is known that the initial process in epithelial corneal wound healing are cell migration with protrusion and polarization [39], and the cell migration decrease in the wound area during the 12 h after corneal epithelial abrasion [40]. After that, cell proliferation activates approximately 24 h after corneal injury, after which tissue remodeling to reestablish the stratified epithelium occurs [41,42]. Thus, the non-wounded area affects the epithelial corneal wound healing until 12 h after abrasion. The wound areas at 0 h in this study (approximately 10.4 mm^2^) were higher than that in previous study (approximately 7.1 mm^2^). Therefore, the different experimental protocols may lead this difference between this and previous results [25,26]. On the other hand, we showed that the substantial shift towards bigger particle sizes occurs with time (Figure 2), although, the therapeutic effect in the N-Ser kept for 1 month prior to treatments was not significantly different in comparison with N-Ser immediately after preparation (*k*_H_ in Wistar rat, 2.29 ± 0.83, *n* = 3). These results showed that the changes of particles size do not significantly affect in the therapeutic effect of N-Ser for 1 month after preparation.

It has been reported that the MAPK/ERK pathways play the regulation of a manifold of physiological and pathological processes [42,43,44,45], and it is well known that ERK1/2, MAPK family members, are also important for cell migration and cell proliferation [46,47,48,49]. In addition, our previous reports showed that sericin enhances cell migration and cell proliferation in the cornea via the phosphorylation of ERK1/2 [19]. Therefore, we measured the changes in ERK1/2 phosphorylation in N-Ser-treated rats. The ERK1/2 phosphorylation increased by the instillation of N-Ser in comparison with sericin (Figure 6). In addition, we showed that the co-instillation of MHN in the N-Ser enhances the intercellular space in the cornea (Figure 3C). From these results, we hypothesize that the co-instillation of MHN increases the ratio of the intercellular space, which may promote better sericin-uptake and -retention in the cornea, and the enhanced sericin levels in the cornea lead to the phosphorylation of ERK1/2, resulting in an increase in epithelial corneal wound healing (Scheme 1). However, the precise mechanism for the changes in the intercellular space by MHN remains unclear. Further studies need in the mechanism for the enhancement of intercellular spacing by MHN, and it is important to clarify the later complications, such as loss of fluid barrier function after the instillation. In addition, it is necessary to demonstrate the mechanism for the increase in pERK1/2 phosphorylation. Recently, Gouveia et al. reported that Yes-associated protein (YAP) is key regulator of pERK in the corneal epithelium, and the enhancement of active YAP lead to increases in pERK in the corneal epithelium [50,51]. Thus, we next plan to demonstrate the effect of MHN in N-Ser on tight junctions in cornea using histological methods. Moreover, we will investigate the relationships the YAP and epithelial corneal wound healing by the sericin treatment.

## 5. Conclusions

We designed an ophthalmic formulation containing sericin and MHN (N-Ser), and found that combination with MHN enhances the therapeutic effect of sericin on corneal wound in comparison with non-combination groups. These findings provide information significant for designing further studies to develop potent drugs to improve severe keratopathy, such as diabetic keratopathy.

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
