# Peer review of "Therapeutic Potential of a Combination of Magnesium Hydroxide Nanoparticles and Sericin for Epithelial Corneal Wound Healing"

_nanomaterials, 2019, doi:10.3390/nano9050768_

Reviewer 1 Report

In the manuscript entitled "Therapeutic Potential of a Combination of Magnesium Hydroxide Nanoparticles and Sericin for Corneal Wound Healing", the authors aim at testing a new therapeutic tool (sericin-carrying magnesium hydroxide nanoparticles) to enhance corneal re-epithelialisation.  The authors are able to show that such nanoparticles do improve corneal epithelial wound closure (albeit only marginally) in both healthy and diabetic rats, without affecting the amount of lacrimal fluid in rabbits.  The mechanisms underlying such effects are then briefly explored.

This is an interesting and timely study that addresses an important concern in the field.  However, in the present form, the article presents several deficiencies that limit its scientific validity and appeal to the readership. As such, a major revision is recommended.

1.     General point: a fresh reading of the article, followed by careful revision must be performed by the authors (and if possible, an English native speaker), as there are several typos, non sequitur sentences, and oddities in the writing.

2.     General point: authors use several abbreviations that are difficult to understand, and not at all defined (e.g., what is SMFC?). Consider using easier nomenclature (e.g. magnesium hydroxide micro-particles = MHP; magnesium hydroxide nanoparticles = MHN; MHN with sericin = N-Ser; MHP with sericin = P-Ser; etc.).

3.     General point: the authors refer throughout the article to (corneal) cells without specifying their type: they should call them epithelial cells, to avoid confusion with other corneal cell types (i.e. stromal or endothelial cells).

4.     Abstract: it is difficult to understand which tests were performed in vivo, ex vivo, and in vitro; authors should make test models clear for each type of assay.

5.     Abstract: abbreviations should be avoided, or at least clearly defined.

6.     Introduction: some citations have improper or misplaced references (e.g., reference 19, 20).

7.     Introduction: the authors refer to (but fail to cite) their previous work (Nagai et al., Biol. Pharm. Bull. 2009, 32:933) on the use of sericin to promote corneal epithelial wound healing in rats. The authors must explain why the current study is substantially different and constitutes an advance over the previous work (especially in light of the apparent much better healing outcome at 12 hours post-injury when using 10% sericin alone!).

8.     Methods: a much greater detail  is needed in sections 2.3 (maybe even using a schematic of photographs of the milling process); 2.4 (Magnesium B test kit protocol); 2.5 (process of repeated instillation, number of animals per condition), 2.6 (how were rats euthanized, how many corneas were used for ex vivo testing of intercellular space in the corneal epithelium, how many images were taken per condition); 2.7 (where were these cells obtained, how many cells were seeded per well, what volumes of Cell Count Reagent SF were used to incubate cells, what wavelength was used to measure reagent absorbance and for calibration; how was the number of cells calculated from the absorbance values); 2.8 (remove results from this section; otherwise similar to notes from 2.5); section 2.9 (how were bands quantified); section 2.10 (wrongly indicated as 4.10) (explain why used standard error instead of the more relevant standard deviation).

9.     Results: Figure 1 and 2 could be merged, with overlapping graphs of particle size immediately after bead preparation and after 1 month incubation, for better comparison. Also, a non-coated (magnesium hydroxide-only) control bead could be included in the analysis. Also, the x-axis scales should be standardised. Also, AFM images should be provided for all conditions. Finally, why did authors use two methods for measuring particle diameter? DLS seems the more accurate, but authors fail to explain the presence of three populations of beads (around 80, 160, and 240 nm), with the bigger ones incidentally increasing in frequency after 1 month incubation, and instead calculate a rough average. This should be explained.

10.  Results: In Figure 3c, micrographs are not detailed enough to show intercellular spacing of the corneal epithelium, and should be replaced by better quality images. In fact, authors should have used immunofluorescence (staining, for example, ZO-1) to more accurately determine this parameter.

11.  Results: In Figure 6, the authors should include a non-treated control. Also, what does the % indicated in the y-axis of the graphs correspond to? Cells? Absorbance? And what does 100% correspond to? Initial number of seeded cells? The values from this analysis are very confusing – it’s just not possible to have more than 100% cell adhesion, it means that the number cells seeded was underestimated, or that untreated HCE-Ts failed to adhere to the plate (which might indicate they were in a poor state), or that cells proliferated during that 12 h period (which is doubtful, and even if occurring, does not correspond to a cell adhesion event). Also, authors should include ERK and pERK immunoblots for all conditions (including non-treated controls).

12.  Discussion: My two main concerns regarding the analysis of these results are:

i) the inconsistency between ERK1/2 activation in vitro and ex vivo. Authors should explain this better. It might be also useful to explain the rationale behind the sericin concentration used in this study (previously, the authors showed that 0.1-0.2% sericin alone produced optimal results in treated HCE-Ts, whereas 10% sericin was optimal in rat corneas).

ii) the argument that increased intercellular space enables better nanoparticle retention and intracellular incorporation of sericin is speculative at this point. Additional experiments showing higher sericin uptake must be performed to support this claim. Moreover, no explanation is given on to why nanoparticles increase intercellular spacing in the corneal epithelium, and if this might lead to later complications (e.g., loss of fluid barrier function).

Author Response

  We carefully revised our manuscript according to the suggestions of the reviewer 1, and details are as follows.

Q1. General point: a fresh reading of the article, followed by careful revision must be performed by the authors (and if possible, an English native speaker), as there are several typos, non sequitur sentences, and oddities in the writing.

A1. In order to respond to the comment, we carefully rechecked the typos, non sequitur sentences, and oddities in the writing in manuscript, and in order to respond to the reviewer 2 comment Q1, we collected several grammatical errors. Thank you very much for pointing this out.

Q2. General point: authors use several abbreviations that are difficult to understand, and not at all defined (e.g., what is SMFC?). Consider using easier nomenclature (e.g. magnesium hydroxide micro-particles = MHP; magnesium hydroxide nanoparticles = MHN; MHN with sericin = N-Ser; MHP with sericin = P-Ser; etc.).

A2. Thank you for pointing out this. In order to respond to the comment, we revised these abbreviations (magnesium hydroxide micro-particles=MHP; magnesium hydroxide nanoparticles= MHN; MHN with sericin=N-Ser; MHP with sericin=P-Ser).

Q3. General point: the authors refer throughout the article to (corneal) cells without specifying their type: they should call them epithelial cells, to avoid confusion with other corneal cell types (i.e. stromal or endothelial cells).

A3. Thank you very much for pointing this out. We revised to epithelial cells. On the other hand, as the response to reviewer 2 comment Q17, we removed the data for HCE-T cells.

Q4. Abstract: it is difficult to understand which tests were performed in vivo, ex vivo, and in vitro; authors should make test models clear for each type of assay.

A4. The reviewer’s comment is correct. In order to respond to the comment, we mentioned the test models clear for each type of assay.

Q5. Abstract: abbreviations should be avoided, or at least clearly defined.

A5. The reviewer’s comments are very important. In order to respond to the reviewer’s comment, we revised the abbreviations in the Abstract.

Q6. Introduction: some citations have improper or misplaced references (e.g., reference 19, 20).

A6. These are excellent points. In order to respond to the reviewer’s comment, we corrected the references (Reference 19 and 20).

Q7. Introduction: the authors refer to (but fail to cite) their previous work (Nagai et al., Biol. Pharm. Bull. 2009, 32:933) on the use of sericin to promote corneal epithelial wound healing in rats. The authors must explain why the current study is substantially different and constitutes an advance over the previous work (especially in light of the apparent much better healing outcome at 12 hours post-injury when using 10% sericin alone!).

A7. It is known that the initial process in corneal wound healing are cell migration with protrusion and polarization, and the cell migration decrease in the wound area during the 12 h after corneal epithelial abrasion. Therefore, the non-wounded area affected the corneal wound healing until 12 h after abrasion. The wound areas at 0 h in this study (approximately 10.4 mm2) were higher than that in previous study (approximately 7.1 mm2), and we think that these difference of the wound area relate. In order to respond to the reviewer’s comment, we added the contents in the Discussion.

Q8. Methods: a much greater detail is needed in sections 2.3 (maybe even using a schematic of photographs of the milling process); 2.4 (Magnesium B test kit protocol); 2.5 (process of repeated instillation, number of animals per condition), 2.6 (how were rats euthanized, how many corneas were used for ex vivo testing of intercellular space in the corneal epithelium, how many images were taken per condition); 2.7 (where were these cells obtained, how many cells were seeded per well, what volumes of Cell Count Reagent SF were used to incubate cells, what wavelength was used to measure reagent absorbance and for calibration; how was the number of cells calculated from the absorbance values); 2.8 (remove results from this section; otherwise similar to notes from 2.5); section 2.9 (how were bands quantified); section 2.10 (wrongly indicated as 4.10) (explain why used standard error instead of the more relevant standard deviation).

A8. Thank you for pointing out this. In order to respond to the reviewer’s comment, we revised following:

    2.3: We added the scheme of preparation process in N-Ser (Figure 1).

    2.4: The samples were mixed with solution containing glucokinase, ATP, glucose and nicotinamide adenine dinucleotide phosphate, and incubated for 3 min, and measured the Abs (340 nm) by UV-2200 (Shimadzu Corp., Kyoto, Japan).

    2.5: The rabbits were instilled twice a day for 1 month (total 60 times). The 5 rabbits/per condition were used in this experiment.

    2.6: The rats were euthanized by the excessive pentobarbital, and 5 samples/group were used for ex vivo testing of intercellular space.

    2.7: As the response to reviewer 2 comment Q17, we removed the data for HCE-T cells. Therefore, we deleted this section.

    2.8: We removed the results from this section, and showed in the result section.

    2.9: The bands were detected using IQuant 400, and the bands were not quantified.

    2.10: We corrected to “2.10”. In the experiment using the animal model, we think that the interval estimation of population mean value need. Therefore, we selected the S.E. in this study.

 Q9. Results: Figure 1 and 2 could be merged, with overlapping graphs of particle size immediately after bead preparation and after 1 month incubation, for better comparison. Also, a non-coated (magnesium hydroxide-only) control bead could be included in the analysis. Also, the x-axis scales should be standardised. Also, AFM images should be provided for all conditions. Finally, why did authors use two methods for measuring particle diameter? DLS seems the more accurate, but authors fail to explain the presence of three populations of beads (around 80, 160, and 240 nm), with the bigger ones incidentally increasing in frequency after 1 month incubation, and instead calculate a rough average. This should be explained.

A9. These are excellent points. We merged the Figs. 1 and 2, and showed as overlapping graphs of particle size immediately after bead preparation and after 1 month incubation (Fig. 2A-C). In addition, we added the data of non-coated MHN (magnesium hydroxide-only). Moreover, we removed the data of nano-size measured by a laser diffraction particle size analyzer SALD-7100, since the accuracy of the data is lower than that in dynamic light scattering NANOSIGHT LM10. The measurement range in NANOSIGHT LM10 is 0-1000 nm. Therefore, the x-axis scales were not changed. Furthermore, in order to respond to the reviewer’s comment, we mentioned the presence of three populations of beads (around 80, 160, and 240 nm), with the bigger ones incidentally increasing in frequency after 1 month incubation, and instead calculate a rough average in the Results.

Q10. Results: In Figure 3c, micrographs are not detailed enough to show intercellular spacing of the corneal epithelium, and should be replaced by better quality images. In fact, authors should have used immunofluorescence (staining, for example, ZO-1) to more accurately determine this parameter.

A10. Thank you for pointing out this. Instead of data used immunofluorescence, we stained the cornea by the fluorescein, and re-analyzed the ratio of the intercellular space in corneal epithelium, and showed as the graph. We think that the result for the intercellular space in corneal epithelium is clear by these revise according to the reviewer comments. Thank you very much for pointing this out. (Figure 3C).

Q11. Results: In Figure 6, the authors should include a non-treated control. Also, what does the % indicated in the y-axis of the graphs correspond to? Cells? Absorbance? And what does 100% correspond to? Initial number of seeded cells? The values from this analysis are very confusing – it’s just not possible to have more than 100% cell adhesion, it means that the number cells seeded was underestimated, or that untreated HCE-Ts failed to adhere to the plate (which might indicate they were in a poor state), or that cells proliferated during that 12 h period (which is doubtful, and even if occurring, does not correspond to a cell adhesion event). Also, authors should include ERK and pERK immunoblots for all conditions (including non-treated controls).

A11. As the response to reviewer 2 comment Q17, we removed the data of HCE-T cells from the Fig. 6. On the other hand, in order to respond to the reviewer’s comment, we added the immunoblots data for the ERK and pERK in the non-treated control. Thank you very much for pointing this out. (Figure 6).

Q12-1. Discussion: the inconsistency between ERK1/2 activation in vitro and ex vivo. Authors should explain this better. It might be also useful to explain the rationale behind the sericin concentration used in this study (previously, the authors showed that 0.1-0.2% sericin alone produced optimal results in treated HCE-Ts, whereas 10% sericin was optimal in rat corneas).

A12. The reviewer’s comment is correct. However, as the response to reviewer 2 comment Q17, we removed the data for HCE-T cells. These revise solved the inconsistency between ERK1/2 activation in vitro and ex vivo in the manuscript.

Q12-2. the argument that increased intercellular space enables better nanoparticle retention and intracellular incorporation of sericin is speculative at this point. Additional experiments showing higher sericin uptake must be performed to support this claim. Moreover, no explanation is given on to why nanoparticles increase intercellular spacing in the corneal epithelium, and if this might lead to later complications (e.g., loss of fluid barrier function).

A12. The reviewer’s comment is correct. In this study, we showed that the enhancement of epithelial corneal wound healing and ERK1/2 activation by the instillation of sericin and N-Ser. Therefore, we think that the sericin uptake increased by the combination of MHN. In addition, we stained the cornea by the fluorescein to discern the area in intercellular and epithelium, and re-analyzed the ratio of the intercellular space in corneas as the additional experiment (Fig. 3C). These results support the relationships of sericin uptake and intercellular space in corneal epithelium. However, further studies need in the mechanism for the enhancement of intercellular spacing by MHN, and, it is important to clarify the later complications, such as loss of fluid barrier function after the instillation. In order to respond to the reviewer’s comment, we added the importance in the Discussion.

Thank you for great comments.

Reviewer 2 Report

The authors described an interesting technique to improve drug (sericin) bioavailability by co-injecting magnesium hydroxide nanoparticles on the ocular surface, with an aim to enhance the epithelial wound healing. The authors also showed the administration of the drug-nanoparticles combination improved the epithelial wound healing in diabetic rats. There are, however, some concerns regarding the grammar and experimental results, which I hope the authors would look into. They are as follows:

1.      Grammatical errors: There are grammatical errors and typos throughout the manuscript. I am not going to list out all of them here, but I sincerely hope the authors would proofread the paper again during revision. Some examples of the errors are as follows:

        -          Line 28: shows. Consider using “showed”.

        -          Line 42: is underwent. Perhaps “occurs” should be used here.

        -          Line 42: is t. Remove “t”.

        -          Line 43: epithelial cells shrinks. It should be “shrink”, as you are describing a plural subject.

        -          Line 44: cell proliferation provide. It should be “provides”, as you are describing a singular subject.

        -          Line 46-47: Not clear. Need to be rephrased.

        -          Line 47-48: In clinically. Should be “Clinically”.

        -          Line 59: are removed. Perhaps the authors would want to consider using “are cleared” here.

        -          Line 60: instillstion. Please correct the spelling.

        -          Line 129: cell-adhesion and cell-growth. Please remove the hyphen in between the two words. Do this for every occurrence of both terms. Also, consider using “cell proliferation” instead of cell growth.

        -          Line 177: 84.6 ± 8.8 nm, 110 ± 4.9 nm. Remove the comma and replace with “and”.

        -          Line 186: sifted. Should be “shifted”.

        -          Line 203: of. Replace it with “the”.

        -          Line 216: woundswere. Please separate the 2 words.

        -          Line 256: cell-function. Please remove the hyphen.

        -          Line 304: display The. Add a full stop in between the 2 words.

        -          Line 322: was not difference in that. Should be “was not different than that in”.

2.      Title: The authors should consider adding “epithelial” in front of corneal wound healing. This is because the manuscript describes only epithelial wound healing.

3.      Line 19: corneal wound healing. Consider adding “epithelial” in front.

4.      Line 20: corneal cell, rabbits, normal rats…. It is confusing. Consider adding words, such as in vivo and in vitro to describe the experiments better.

5.      Line 25: expand the intercellular space in the cornea. Is it intercellular or intracellular? It appears the authors used the term interchangeably in the manuscript. For example, line 171, line 296, and line 344.

6.      Line 63: the ratio of the intercellular space. The ratio in relation to what? In relation to normal epithelium? This term should be made clear here, because it is used repeatedly throughout the manuscript.

7.      Line 71-74: Can the authors mention how many in total of each animal was used in the methods section?

8.      Line 120: twice a day. Is there a particular reason in the discrepancy of the instillation frequency between the rabbits and rats (5 times a day)? Please explain so that the readers understand the rationale of the methods.

9.      Line 179: nano-size. Can the authors also describe the shape? Is it spherical or rod shape?

10.  Line 183: there is no significant difference. The P value should be stated. This applies for any statement regarding significance or non-significance of comparisons that the authors made throughout the manuscript.

11.  Particle size: There appears to be a slight increase in the particle size following 1-month storage. The authors claimed that there was no agglomeration of nanoparticles. Can the authors supplement the claim with a polydispersity index (PdI) in the manuscript? I am sure the PdI is given in every DLS measurement.

12.  Figure 3C: It is extremely difficult to differentiate and distinguish the intercellular space in the images. Ideally, this could be resolved by staining with CK3/12 to differentiate the space and epithelial cells. I strongly suggest the authors to do so, in order to support the claim better. Other alternative method deemed better by the authors is also welcome.

13.  Line 241: n=6-12. Which group had 6, 7, 8,… or 12 animals? It is better to state the number of animals for each group to make it clearer.

14.  Line 249: n=6-10. See point #13 above.

15.  Line 255: n=6-12. See point #13 above.

16.  Line 343: in vitro or in vivo?

17.  Figure 6: I am not sure what the purpose of presenting the HCET cell results here is. Perhaps it can be excluded from the paper. First, this is the only part of the study that used HCET cells and it has no correlation with the rest of the study. Second, the experiments do not show any difference anyway. If the authors think it is important to be included in the paper, please elaborate the significance of the HCET cell results in the discussion. Also elaborate a bit more, why there is a difference in the p-ERK1/2 between HCET and in vivo rats.

18.  The fate of the nanoparticles: Can the authors comment on the fate of the nanoparticles after the wound closure? Do the nanoparticles stay in between the cells or do they get sloughed off as the cells undergo Y and Z differentiation process? Or do the nanoparticles biodegrade over time? Can the nanoparticles become more soluble over time in the lacrimal fluid?

Author Response

   We carefully revised our manuscript according to the suggestions of the reviewer 2, and details are as follows.

Q1. Grammatical errors: There are grammatical errors and typos throughout the manuscript. I am not going to list out all of them here, but I sincerely hope the authors would proofread the paper again during revision.

A1. In order to respond to the reviewer’s comment, we collected these all grammatical errors, and carefully rechecked in the manuscript. Thank you for pointing out this.

Q2. Title: The authors should consider adding “epithelial” in front of corneal wound healing. This is because the manuscript describes only epithelial wound healing.

A2. Thank you very much for pointing this out. In order to respond to the reviewer’s comment, we changed the title to “Therapeutic Potential of a Combination of Magnesium Hydroxide Nanoparticles and Sericin for Epithelial Corneal Wound Healing”.

Q3. Line 19: corneal wound healing. Consider adding “epithelial” in front.

A3. These are excellent points. In order to respond to the reviewer’s comment, we addedepithelial” in front of corneal wound healing.

Q4. Line 20: corneal cell, rabbits, normal rats…. It is confusing. Consider adding words, such as in vivo and in vitro to describe the experiments better.

A4. The reviewer’s comment is correct. In order to respond to the reviewer’s comment, we revised the sentence.

Q5. Line 25: expand the intercellular space in the cornea. Is it intercellular or intracellular? It appears the authors used the term interchangeably in the manuscript. For example, line 171, line 296, and line 344.

A5. Thank you for pointing out this. In order to respond to the reviewer’s comment, we revised to “intercellular”.

Q6. Line 63: the ratio of the intercellular space. The ratio in relation to what? In relation to normal epithelium? This term should be made clear here, because it is used repeatedly throughout the manuscript.

A6. The reviewer’s comments are very important. The “the ratio of the intercellular space in cornea” show “the area in intercellular / area in intercellular and epithelium”. In order to respond to the reviewer’s comment, we mentioned the definition.

Q7. Line 71-74: Can the authors mention how many in total of each animal was used in the methods section?

A7. Thank you very much for pointing this out. In order to respond to the reviewer’s comment, we added the number (Japanese albino rabbits, n=8; Wistar rats, n=55; OLETF rats, n=31) in the methods section.

Q8. Line 120: twice a day. Is there a particular reason in the discrepancy of the instillation frequency between the rabbits and rats (5 times a day)? Please explain so that the readers understand the rationale of the methods.

A8. The reviewer’s comment is correct. The rat with corneal abrasion is acute model. Therefore, we performed the multi instillation (5 times a day). On the other hand, the normal rabbit is used in study for long-term administration, and applied the general instillation (2 times a day). In order to respond to the reviewer’s comment, we added the information in the Materials and Methods.

Q9. Line 179: nano-size. Can the authors also describe the shape? Is it spherical or rod shape?

A9. These are excellent points. In the AFM image, the shape was like spherical shape. In order to respond to the reviewer’s comment, we mentioned the content in the Results.

Q10. Line 183: there is no significant difference. The P value should be stated. This applies for any statement regarding significance or non-significance of comparisons that the authors made throughout the manuscript.

A10. The comment is correct. We corrected the sentence to “the presence of three populations of beads (around 80, 160, and 240 nm) were observed, although the MH particles in N-Ser were still in the nano-order size range, and the shape of the MH particles 1 month after preparation was also similar to that immediately after preparation”.

Q11. Particle size: There appears to be a slight increase in the particle size following 1-month storage. The authors claimed that there was no agglomeration of nanoparticles. Can the authors supplement the claim with a polydispersity index (PdI) in the manuscript? I am sure the PdI is given in every DLS measurement.

A11. These are excellent points. We measured the particle size by the NANOSIGHT LM10. Although, the NANOSIGHT LM10 is DLS method, the system can not calculate the polydispersity index (PdI), since the measurement principle (nanoparticle tracking analysis) is different from Photon Correlation Spectroscopy and frequency analysis. Instead of the PdI, we revised the sentence following the reviewer’s comment Q10.

Q12. Figure 3C: It is extremely difficult to differentiate and distinguish the intercellular space in the images. Ideally, this could be resolved by staining with CK3/12 to differentiate the space and epithelial cells. I strongly suggest the authors to do so, in order to support the claim better. Other alternative method deemed better by the authors is also welcome.

A12. The reviewer’s comment is correct. We stained the cornea by the fluorescein, and re-analyzed the ratio of the intercellular space in corneal epithelium, and showed as the graph. The result for the intercellular space in corneal epithelium is clear by the revise according to the reviewer comments. Thank you very much for pointing this out. (Figure 3C).

Q13-15. Line 241: n=6-12. Line 249: n=6-10. Line 255: n=6-12. Which group had 6, 7, 8,… or 12 animals? It is better to state the number of animals for each group to make it clearer.

A13-15. The reviewer’s comments are very important. In order to respond to the reviewer’s comment, we added each number in the Fig legends and Table.

Q16. Line 343: in vitro or in vivo?

A16. In this revise process, the sentence was deleted. Thank you very much for pointing this out.

Q17. Figure 6: I am not sure what the purpose of presenting the HCET cell results here is. Perhaps it can be excluded from the paper. First, this is the only part of the study that used HCET cells and it has no correlation with the rest of the study. Second, the experiments do not show any difference anyway. If the authors think it is important to be included in the paper, please elaborate the significance of the HCET cell results in the discussion. Also elaborate a bit more, why there is a difference in the p-ERK1/2 between HCET and in vivo rats.

A17. The reviewer’s comment is correct. In order to respond to the reviewer’s comment, we removed the data for HCET cell (Figure 6).

Q18. The fate of the nanoparticles: Can the authors comment on the fate of the nanoparticles after the wound closure? Do the nanoparticles stay in between the cells or do they get sloughed off as the cells undergo Y and Z differentiation process? Or do the nanoparticles biodegrade over time? Can the nanoparticles become more soluble over time in the lacrimal.

A18. Thank you very much for pointing this out. We previously reported that the Mg2+ levels in the aqueous humor were enhanced by the instillation of solution containing Mg2+. In contrast of the results in solution containing Mg2+, the MHN did not penetrate into the aqueous humor, since no change in the Mg2+ levels in the aqueous humor of rabbits treated with MHN dispersions were observed. These findings suggested that almost all of the MHN were delivered to the stomach via the nasolacrimal duct after instillation. In order to respond to the comment, we added the information in the Discussion.

Thank you for great comments.

Round  2

Reviewer 1 Report

The authors addressed most of the previous comments to my satisfaction. However, some issues remain:

1) Abstract and Introduction - the writing requires serious proof-reading and polishing (preferably by english-native speaker).

2) Results, section 3.1 - I do not agree with the authors' analysis presented in Fig. 2. It is rather obvious that MHN and N-Ser occur in three distinct populations of particle sizes (around 80, 160, and 240 nm), and that a substantial shift towards bigger particle sizes occurs with time. Instead, the authors calculate an average size (!) instead of presenting a percentage for each particle population, and associate a tiny variation despite the very considerable range (misleading the reader). How was this calculated? But more importantly, how does this shift impact on particle function? If, as authors suggest, particle function depends on cel permeation, wouldn't they expect lower function with increased particle size? Do the authors explored the effects of N-Ser kept for 1 month prior to treatments? This is incredibly important for the usefulness of MHNs as off-the-shelf drug delivery system.

3) Results, section 3.3 - authors should provide a quantification of the levels of pERK based on band intensity, and with appropriate control normalization.

4) Discussion - the authors suggest that the increased migration rates of the rat epithelial cells treated with sericin were due to activation of pERK; it would be interesting if they could briefly discuss how this occurs - for example, due to its 'glue' properties, sericin may be locally increasing cell and/or tissue biomechanics, which according to very recent findings (see https://doi.org/10.1038/s41467-019-09331-6 and https://doi.org/10.3390/cells8040347) is instrumental to the promotion of corneal epithelial cell migration (via YAP activation). YAP is also a know regulator of pERK, with increases in active YAP leading to increases in pERK. I suggest that discussing this possible mechanism, along with its possible inclusion in future studies, will broaden the scope - and the impact - of the manuscript.

Author Response

We carefully revised our manuscript according to the suggestions of the reviewer 1, and details are as follows.

Q1. Abstract and Introduction - the writing requires serious proof-reading and polishing (preferably by english-native speaker).

A1. In order to respond to the comment, we requested an English check by a native English-speaking person with sufficient scientific knowledge (Dr. Margaret Dooley Ohto).

Q2. Results, section 3.1 - I do not agree with the authors' analysis presented in Fig. 2. It is rather obvious that MHN and N-Ser occur in three distinct populations of particle sizes (around 80, 160, and 240 nm), and that a substantial shift towards bigger particle sizes occurs with time. Instead, the authors calculate an average size (!) instead of presenting a percentage for each particle population, and associate a tiny variation despite the very considerable range (misleading the reader). How was this calculated? But more importantly, how does this shift impact on particle function? If, as authors suggest, particle function depends on cel permeation, wouldn't they expect lower function with increased particle size? Do the authors explored the effects of N-Ser kept for 1 month prior to treatments? This is incredibly important for the usefulness of MHNs as off-the-shelf drug delivery system.

A2. Thank you for pointing out this. The particle size was calculated using computerized image analysis software connected to the NanoSight LM10 system. The particle size was re-checked, and confirmed that the particle size were right. We think that the substantial shift towards bigger particle sizes occurs with time were reflected in the S.D. In addition, we have the data for experiment by using N-Ser kept for 1 month prior to treatments. The therapeutic effect in the N-Ser kept for 1 month prior to treatments was not significantly different in comparison with the result using N-Ser immediately after preparation. These results showed that the particles in the N-Ser shift towards bigger particle sizes occurs with time (1 month), however the changes of particles size don’t significantly affect in the therapeutic effect of N-Ser. In order to respond to the comment, we added the contents in the Discussion.

Q3. Results, section 3.3 - authors should provide a quantification of the levels of pERK based on band intensity, and with appropriate control normalization.

A3. Thank you very much for pointing this out. In order to respond to the comment, we calculated the band intensity, and added the data in the Results.

Q4. Discussion - the authors suggest that the increased migration rates of the rat epithelial cells treated with sericin were due to activation of pERK; it would be interesting if they could briefly discuss how this occurs - for example, due to its 'glue' properties, sericin may be locally increasing cell and/or tissue biomechanics, which according to very recent findings (see https://doi.org/10.1038/s41467-019-09331-6 and https://doi.org/10.3390/cells8040347) is instrumental to the promotion of corneal epithelial cell migration (via YAP activation). YAP is also a know regulator of pERK, with increases in active YAP leading to increases in pERK. I suggest that discussing this possible mechanism, along with its possible inclusion in future studies, will broaden the scope - and the impact - of the manuscript.

A4. The reviewer’s comment is correct. In order to respond to the comment, we added the possible mechanism and cited the 2 references in the Discussion (References 50 and 51).

 Thank you for great comments.

Reviewer 2 Report

I have no further comments for the authors.

Author Response

Thank you for great comments.